# Identification of miRNAs Mediating Seed Storability of Maize during Germination Stage by High-Throughput Sequencing, Transcriptome and Degradome Sequencing

**DOI:** 10.3390/ijms232012339

**Published:** 2022-10-15

**Authors:** Yongfeng Song, Zhichao Lv, Yue Wang, Chunxiang Li, Yue Jia, Yong Zhu, Mengna Cao, Yu Zhou, Xing Zeng, Zhenhua Wang, Lin Zhang, Hong Di

**Affiliations:** Key Laboratory of Germplasm Enhancement, Physiology and Ecology of Food Crops in Cold Region, Engineering Technology Research Center of Maize Germplasm Resources Innovation on Cold Land of Heilongjiang Province, Northeast Agricultural University, Harbin 150030, China

**Keywords:** maize, seed storability, miRNA, degradome sequencing, transcriptome

## Abstract

Seed storability is an important trait for improving grain quality and germplasm conservation, but little is known about the regulatory mechanisms and gene networks involved. MicroRNAs (miRNAs) are small non-coding RNAs regulating the translation and accumulation of their target mRNAs by means of sequence complementarity and have recently emerged as critical regulators of seed germination. Here, we used the germinating embryos of two maize inbred lines with significant differences in seed storability to identify the miRNAs and target genes involved. We identified a total of 218 previously known and 448 novel miRNAs by miRNA sequencing and degradome analysis, of which 27 known and 11 newly predicted miRNAs are differentially expressed in two maize inbred lines, as measured by Gene Ontology (GO) enrichment analysis. We then combined transcriptome sequencing and real-time quantitative polymerase chain reaction (RT-PCR) to screen and confirm six pairs of differentially expressed miRNAs associated with seed storability, along with their negative regulatory target genes. The enrichment analysis suggested that the miRNAs/target gene mediation of seed storability occurs via the ethylene activation signaling pathway, hormone synthesis and signal transduction, as well as plant organ morphogenesis. Our results should help elucidate the mechanisms through which miRNAs are involved in seed storability in maize.

## 1. Introduction

Seed storability is critical to ensure seed safety and vitality [1]. Generally, with the extension of aging time, the color of the seed surface darkens, the germination rate decreases, and seedling growth in the field is inhibited. A series of harmful cell metabolism activities occur inside the seeds, whereby different biochemical changes, such as lipid peroxidation, enzyme inactivation, cell membrane damage, decreased energy production capacity, proteins, DNA and RNA degradation, or the accumulation of reactive oxygen species, can lead to a decline in seed storability and affect germination rates [2,3,4]. For example, the seed vigor of soybean, rice, Arabidopsis, and other seeds decreases with aging time, leading to a decrease in the seed germination rate and germination potential, and inhibiting the growth of young buds and roots [5,6]. Previous studies in rice, *Arabidopsis*, and maize showed that seed storability is a quantitative trait controlled by multiple genes [7,8], whereby QTL analysis provides a powerful tool for the study of this trait [9,10,11]. In rice, wheat, and *Arabidopsis*, the lipid oxidase (*LOX*) gene affects seed storability by catalyzing the oxidation of unsaturated fatty acids through the linoleic acid to produce free radicals and lipid peroxides [12,13,14], whereby depleted or low levels of *LOX* improve seed storability [15]. The acetaldehyde dehydrogenase (*ALDH*) gene family catalyzes the oxidation of aldehydes into corresponding carboxylic acids. For example, *OsALDH7* can improve seed storability due to its strong activity against malondialdehyde and ability to remove aldehydes produced by lipid peroxidation during seed drying [16]. Moreover, the overexpression of *ZmGOLS2* and *ZmRS* has been shown to increase seed storability in *Arabidopsis* [17]. Likewise, the basic α -galactosidase 1 (*ZmAG1*) can be translated and induced during maize seed germination, hydrolyze the raffinose family oligosaccharides (RFO), and regulate seed storability and germination by regulating the expression of *ZmAGA1* [18]. However, the specific regulatory mechanisms involved in seed storability in maize remain unclear and relatively understudied.

MicroRNAs (miRNAs) can regulate various physiological processes, such as development, growth, or resistance to stress [19]. The development of degradome sequencing techniques, which is a high-throughput strategy based on the parallel analysis of RNA ends (PARE), has successfully helped to identify novel miRNAs and respective target genes [20] in order to evaluate miRNA self-regulation [21] and determine the relationship between miRNAs and its target genes [22]. Previous studies indicated several miRNAs are involved in seed development, such as miR156, miR172, miR402, and miR398b [23,24]. In addition, several mutations on genes essential for small RNA synthesis, such as *AGO1*, *HEN1*, *DCL1*, and *HYL1*, caused serious defects in seed development and embryogenesis [25]. MiRNAs are essential for plant growth and development by participating in plant hormone signal transduction pathways and regulating interactions with the plant’s environment [26]. In *Arabidopsis*, miR159 and the target genes *MYB33* and *MYB101* are crucial for seed germination through the abscisic acid (ABA) pathway [27]. In addition, this miRNA participates in GA signal transduction by regulating the *GAMYB* gene [28], and its expression can be activated by the presence of ethylene in the root [29], which may explain the association between ethylene and ABA during seed germination. Other miRNAs participate in seed germination, such as miR160, which affects the auxin signaling pathway by clipping the target genes *ARF10* and *ARF16*, and regulates the expression of *ABI3*, an upstream gene of the ABA signaling pathway, thereby controlling seed dormancy and affecting germination and post-embryonic development [30]. With the development of seed morphology, stored proteins, sugars, starch, and lipids accumulate continuously, playing an important role in late seed dormancy and germination. MiR1211 participates in carbohydrate metabolism and impacts the accumulation of stored substances in seeds by regulating *SnRK1* (Snf1 related protein kinase regulatory subunit beta-1) [31]. In rice, the up-regulation of miR164c resulted in decreased seed storability, while the up-regulation of this miRNA benefits the maintenance of seed vigor [5]. Additionally, miR164c’s target genes *OsPSK5* and *OMTN2* affect *RPS27AA*, a key gene regulating rice seed storability through energy metabolism, endoplasmic reticulum stress response, and embryonic development [32]. Few studies to date have reported the relationship between miRNAs and seed development in maize and have mostly focused on endosperm development and embryogenesis. Gu et al. (2013) identified miRNAs during maize endosperm development and obtained 95 known miRNAs and 18 putative miRNAs. In maize somatic embryogenesis, miR528 regulates multi-target mRNA in somatic embryogenesis (SE) by promoting the degradation of multi-target mRNA and/or translation inhibition. *MATE*, *bHLH*, and *SOD1a* are negatively correlated with miR528 [33]. Moreover, miR166 and its target gene *HD-ZIP III* (class III homeodomain leucine zipper) regulate seed maturation and lateral root growth [34,35,36], while miR156, miR164, miR167, miR168, miR169, and miR396 are involved in the regulation of heterosis during seed germination [37]. In addition, miR160 and miR167 are, respectively, down- and up-regulated at the late stages of endosperm development, impacting this process by targeting *ARF* [38]. During the development process of seeds in two maize inbred lines (PH4CV and PH6WC), the differentially expressed miR156, miR171, miR396, and miR444, which, respectively, target *SPL*, *SCL*, *QQT*, and *EDA*, regulate flowering and embryonic development [39]. In addition, miR319a-3p and its target genes *MYB33* and *MYB101* participate in the regulation of seed storability in sweet maize through the MYB protein-binding chromatin pathway [40].

Gong et al. (2015) assessed seed storability in sweet corn seeds and found that the expression of zma-miR319a-3p_R+1 displayed an up-regulating trend as sweet corn seed viability declined, indicating that miR319 had an inhibitory effect on seed viability [40]. Additionally, miR319a-3p negatively regulates *MYB33* and *MYB101*, which are involved in seed storability. Here we adopted the method of high temperature and humidity artificial aging in two maize inbred lines with significant differences in seed storability and performed global RNA analysis at 24 h of seed imbibition after 4 days of aging. In the analysis we included transcriptome, in addition to miRNA and degradome sequencing, to identify differentially expressed miRNAs and their target genes more accurately. The results of qRT-PCR and 5’RLM-RACE showed that 5 miRNAs were down-regulated and 1 miRNA was up-regulated with the decrease of seed vigor. GO and KEGG bioinformatic analyses revealed that these miRNA and their negative target genes were involved in regulating the ethylene activation signal pathway, hormone synthesis, signal transduction, and plant organ morphogenesis, thereby affecting seed storability. These results provide crucial information to the understanding of the regulatory pathways and mechanisms through which miRNA and its target genes regulate seed storability in maize.

## 2. Results

### 2.1. The Artificial Aging and Sampling Method of Inbred Lines

The aging time of seeds and the sampling time at the germination stage are very important to evaluate the test results. The germination percentage of Dong156 and Dong237 was 98% and 95% under normal circumstances, respectively. Following artificial aging at high temperature and humidity (45 °C, 95% RH), the germination rate of seeds were declining. After 10 days of artificial aging, the germination rate of Dong237 decreased to 0, and after 16 days of artificial aging, the germination rate of Dong156 approached 0 (Appendix A). The two inbred lines showed extremely significant differences between 3 and 4 days of artificial aging. Their germination rates were 82% and 53%, respectively, after 4 days of aging treatment, which was more significant than 3 days comparisons (*p* < 0.01). This helped us to determine that 4 days of artificial aging was the appropriate treatment time for this experiment (Figure 1A). We determined the seed water content at different germination times by soaking seeds in a paper bed and, following changes in water content, divided seed germination into the rapid water absorption stage, the stagnant water absorption stage, and the second rapid water absorption stage. The inflection points were established at 6 h, 14 h, 24 h, 30 h, and 40 h, respectively. The seeds transitioned from the water absorption stagnation stage to the second rapid water absorption stage 24 h after germination, at which point the seeds basically completed absorption and swelling, presented a smooth surface, the radicle position began to protrude, the water content remained relatively constant, and the metabolic activity increased along with the transcription of many new genes (Figure 1B,C).

### 2.2. Overview and Validation of Small RNA Sequencing Data

To study miRNAs that may be related to seed storability in maize seeds during germination, we analyzed the expression of miRNAs during the artificial aging of seeds. A total of six maize small RNA libraries were prepared from embryonic tissues of Dong156 and Dong237 24 h after artificial aging germination. Deep sequencing generated 13.45, 13.72, and 14.07 million original sequences, respectively, from the artificial aging Dong156 banks termed AA156-24h1, AA156-24h2, and AA156-24h3 and 13.83, 14.16, and 14.00 million original sequences, respectively, from the artificial aging Dong237 banks termed AA237-24h1, AA237-24h2, and AA237-24h3. After deleting the sequences for which adapters could be found, and those with length <18 or >25 nucleotides, a total of 5.50, 5.69, and 8.44 million clean reads were obtained from the AA156-24h1, AA156-24h2, and AA156-24h3 libraries and kept for further analysis, respectively. Analogously, we retained a total of 8.06, 4.41, and 3.13 million clean reads, respectively, for the AA237-24h1, AA237-24h2, and AA237-24h3 libraries (Appendix A). Unique sequences with lengths of 18–25 nucleotides were mapped to the miRBase 22.0 (ftp://mirbase.org/pub/miRBase/CURRENT/ (accessed on 15 May 2022)). Among them, we found that the sequence length of most small RNAs in the six libraries was 21 nt or 24 nt, of which 21 nt accounted for 20.89% and 24 nt for 54.32% (Figure 2A). The difference significance was calculated and analyzed using the expression frequency in the high-throughput sequencing results (Figure 2B).

To identify miRNAs in our dataset, we conducted a quick search between the sequences identified by deep sequencing and miRNAs from mature plants currently deposited in the miRBase database and identified the known miRNAs and novel miRNA sequences derived from 3′ (3p in miRNA names) and 5′ (5p in miRNA names) ends. This allowed us to identify 218 known miRNAs from 29 known families, such as miR156, miR160, miR166, miR167, miR168, miR169, miR396, and miR444, of those identified the miR166 family has the largest number (Figure 2C), followed by the miR156 family (with 22 members), and miR169 and miR396 with more than 10 members each. In comparison, in the database gene expression group AA156VSAA237, we found 27 known miRNAs differentially expressed, of which 13 were up-regulated and 14 down-regulated (Table 1). The sequencing reads that could not be mapped to the miRBase were used for searches against the Rfam, the mRNA, and the repeat Repbase databases. After removing the mapped reads, we predicted the potential dry ring structure of the remaining reads. A total of 448 new miRNA molecules with 20–24 nt in length were obtained by filtering out miRNAs with low expression levels (77.87% of the total molecules were 24 nt long, with a lower proportion of 18–19 nt sequence). In the two inbred lines of the AA156VSAA237 library, we found only 20 putative miRNAs (named novel-miR1-20) with at least a 1.5-fold expression change, *p*-value < 0.05 (Appendix A).

In order to confirm the accuracy of high-throughput sequencing, we selected six differentially expressed miRNAs and verified the expression level of miRNAs by using qRT-PCR. The results showed that among these miRNAs selected, zma-miR169o-5p, zma-miR390a-5p, zma-miR396c_L-1, zma-miR397b-p5, and novel-miR4 were down-regulated with the decrease of seed vigor, and the expression of zma-miR444 was up-regulated with decreasing seed vigor, which was consistent with the expression obtained by high-throughput sequencing (Figure 3A,B).

### 2.3. Prediction of Target Genes of miRNAs by Degradome Sequencing and Validation of miRNA Targets by 5′RLM-RACE

We performed the Degradome sequencing of Dong156 and Dong237, which, respectively, generated 1.15 and 1.21 million original readings, as well as 0.32and 0.30 million unique original readings for miRNA target gene analysis. The target genes of all miRNAs with differential significant expression (*p* ≤ 0.05) were verified by the plant miRNA target Gene prediction software TargetFinder and degradation group sequencing, and the functions and biological processes of the identified miRNA target genes were analyzed by Gene Ontology (GO). In comparison, in group AA156VSAA237, 133 negatively regulated target genes corresponding to 25 known differentially expressed miRNAs were screened out (Appendix A), as well as 11 newly predicted differentially expressed miRNAs and 27 negative regulation target genes (Appendix A). Combined with GO enrichment analysis, we identified 13 pairs of negatively regulated target genes corresponding to miRNAs that were highly correlated with seed storability, nine miRNAs were down-regulated (zma-miR156e-3p, zma-miR169o-5p, zma-miR398b-5p, zma-miR390a-5p, zma-miR396c, zma-miR408a, zma-miR397b-p5, novel-miR1, and novel-miR12), and four miRNAs that were up regulated (zma-miR319d-p5 and zma-miR444a, novel-miR4 and novel-miR6) (Figure 4). These miRNAs were then analyzed using thermographic clustering (Figure 5A). The enriched GO functions associated include seed development, transcriptional regulation, seed germination, auxin mediated signal pathway, abscisic acid activated signal pathway, embryonic development ending with seed dormancy, and ethylene activated signal pathway, the biological processes AMP deaminase activity, DNA binding transcription factor activity, transcription factor binding and DNA binding in the transcription regulatory region, and other molecular functional processes (Figure 5B). In addition, we found that genes related to purine metabolism and phytohormone signaling were enriched in the KEGG analysis (Figure 5C). These results suggest these target genes play a key role in maize seed storability.

In addition, in order to confirm the accuracy of degradome sequencing, we selected eight differentially expressed target genes, and verified the expression level of these target genes by using qRT-PCR. The results showed that among the eight target genes selected, *GRMZM2G165488*, *GRMZM134396*, *GRMZM2G105335*, *GRMMZM2G09894*, *GRMZ M2G133568*, and *GRMZM2G009871* were up-regulated with the decrease of seed vigor, while *GRMZM2G399072* was down-regulated with the decrease of seed vigor. This trend was consistent with degradome sequencing but only differed in numerical value. Combined with the results of differential expression miRNAs verified by small RNA sequencing, we found that zma-miR169o-5p, zma-miR390a-5p, zma-miR396c_L-1, zma-miR396c_L-1, zma-miR397b-p5, and novel-miR4 expression was negatively related to the expression of its target genes, respectively, the up-regulated expression of zma miR444 was negatively correlated with the target gene *GRMZM2G399072*, which is consistent with the functional characteristics of miRNA in plants. We noted that *GRMZM2G165488* is the target gene of zma-miR169o-5p, *GRMZM2G134396* is the target gene of zma-miR390a-5p; *GRMZM2G105335*, *GRMZM3G098594*, and *GRMZM2G133568* are the target genes of zma-miR396c_L-1; *GRMZM2G399072* is the target gene of zma-miR444; and *GRMZM2G009871* is the target gene of novel-miR4 (Figure 3C,D). The degradome sequencing analysis predicted eight targets that might be cleaved by seven miRNAs according to the relative abundance of target mRNA site markers (Figure 3E–L). The target positions of zma-miR169o-5p, zma-miR390a-5p, zma-miR396c, zma-miR397b-p5, zma-miR444, and novel-miR4 are 1504, 1451, 649, 605, 764, 10791, 2080, and 1385, respectively.

We selected two interested zma-miR169o-5p and zma-miR444a to verify whether they promote the cutting of selected targets, 5′RLM-RACE amplification was performed followed by the mapping of the cleavage sites to their corresponding transcripts. According to the gene sequence information and miRNA binding sites, the cleavage site was predicted, and combined with the peak map of cleavage site sequencing, we found that the action site of zma-miR169o-5p_R-1 on the target gene *GRMZM2G165488* was in the 3’UTR region. The action site of zma-miR444a_1ss12TC on the target gene *GRMZM2G399072* was in exon 9. The 5 ‘RACE results further confirmed that zma-miR169o-5p_R-1 and zma-miR444a_1ss12TC mediated the splicing of corresponding target genes (Figure 3M,N and Appendix A).

### 2.4. Transcriptome Analysis Explored the of DEGs Involved in Maize Seed Storability

The GO-LoopCircos.Q of transcriptome sequencing revealed that DEGs are involved in a variety of biological processes (“transcriptional regulation”, “protein phosphorylation”, “REDOX processes”, “response to abscisic acid”, “embryonic development ending with seed dormancy”, “abscisic acid activated signaling pathway”, and “ethylene activated signaling pathway”), cellular components (“ nucleus “, “cytoplasm”, “plasma membrane”, “membrane components”, and “mitochondria”), and molecular functions (“ protein binding “, “DNA-binding transcription factor activity”, “ATP binding”, “protein serine/threonine kinase activity”, “ubiquitin protein ligase activity”, and “calcium binding activity”). It is worth noting that the “binding” function accounts for most of the rich paths (Figure 6A) (Appendix A). The overall distribution of differentially expressed genes (DEGs) can be visualized using a volcano map (Figure 6B). A total of 13,353 DEGs (|log2FC| ≥ 0.585, *p* < 0.05) were detected in the AA156-24h and AA237-24h comparisons, of which 6585 were up-regulated while 6768 were down-regulated (Figure 6C). Hierarchical cluster analysis showed that a higher number of genes were down-regulated in the artificial aging treatment group (AA156VSAA237). KEGG analysis indicated that DEGs are associated with “plant hormone signal transduction”, “plant MAPK signal pathway”, “starch and sucrose metabolism”, and “plant pathogen interactions” (Figure 6D) (Appendix A). Once again, our results indicated that these DEGs are involved in seed storability.

### 2.5. The Combined Analysis of Transcriptome, Small RNA, and the Degradome Sequencing

The transcriptome and miRNA of maize seeds significantly changed after artificial aging. The analysis of these miRNA degradomes and transcriptomes indicated that most genes with altered expressions were enriched in GO terms related to the nucleus, regulation of transcription and DNA templates, and abscisic acid signaling pathway (Appendix A). To detect negatively regulated miRNAs and their target genes, the combined analysis of the expression of miRNAs, the degradome, and the transcriptome was performed. We identified a total of eight negatively regulated genes (*GRMZM2G165488*, *GRMZM2G134396*, *GRMZM2G133568*, *GRMZM2G098594*, *GRMZM2G105335*, *GRMZM2G033230*, *GRMZM2G399072*, and *GRMZM2G009871*) targeted by six miRNAs (zma-miR169o-5p, zma-miR390a-5p, zma-miR396c, zma-miR397b-p5, zma-miR444, and novel-miR4) (Table 2). The expression changes in these genes were higher than 1.5-fold (|log2FC| > 0.58, *p* < 0.05). These miRNAs and target gene pairs are involved in multiple biological processes, including the ethylene signaling pathway (miR444a-5p-*AP2-ERF*), the abscisic acid signaling pathway pmiR169o-5p -*NF-YA11* and miR390o-5p-AMP deaminase), Gibberellin signaling (miR396c-*GRF*), or ATP binding (novel-miR4-*TCP*-*1*). In particular, *GRMZM2G165488* is negatively regulated by miR169o-5p and associated with three GO categories: GO:0016602 (CCAAT-binding factor complex), GO:0009793 (embryo development ending in seed dormancy), and GO:0006355 (regulation of transcription, DNA-templated). The four DEGs of GO:0016602 are transcription factors of the NF-YA family, the six DEGs of GO:0009793 are annotated as *NF-YA*, *NAC*, and *b-ZIP*. The 27 DEGs of GO:0006355 mainly include *AP2-ERF*, *bZIP*, *NF-YA*, and *GRF*. Of these, *AP2-ERF* is negatively regulated by novel-miR4 that is associated with GO:0007275 (multicellular biological development), GO:0009873 (ethylene activation signaling pathway), and GO:1905392 (plant organ morphogenesis). Most of the 15 DEGs detected by these GO terms belong to the AP2-ERF and NAC families. Finally, other detected DEGs were mostly related to GO:0009737, GO:0000977, and GO:0009793 (Appendix A).

## 3. Discussion

We found that miR164 is closely related to seed storability in rice; the seeds of the miR164c silencing line (MIM164c), and the miR164c overexpression line (OE164c) have significant differences in antiaging ability and gene and protein expression levels. Differential expression genes (DEG) or proteins are significantly enriched in “endoplasmic reticulum protein processing”, “embryonic development”, “energy metabolism”, and other aspects related to seed vitality [32]. The GO enrichment analysis of miRNAs corresponding to target genes found in our research also includes “embryonic development” and “endoplasmic reticulum protein processing”. Our research found that miR169o-5p was down-regulated in AA156VSAA237, indicating that miR169o-5p was significantly expressed in Dong237, which had poor storability and inhibited seed vigor. We speculated that the seeds of miR169o-5p silencing line (MIM164o-5p) and miR169 overexpression line (OE169o-5p) will have significant differences in antiaging ability, which needs to be verified in future experiments. In addition, we also focused on hormone-related pathways, such as “response to abscisic acid” and “ethylene signaling pathway”, which are closely related to seed germination and development. The overall inhibition of miRNA in hybrids may lead to increased gene expression, which may be one of the reasons why hybrids show higher embryonic germination vitality than their parents [37]. Gong et al. (2015) assessed seed storability in sweet corn seeds and found that the expression of zma-miR319a-3p_R+1 showed an up-regulated trend as sweet corn seed viability declined, indicating that miR319 had an inhibitory effect on seed viability [40], by miR319a-3p negatively regulating *MYB33* and *MYB101*, which are involved in seed storability. The zma-miR319a-3p_R+1 was not identified in our study, which could be due to the materials used and the sampling time. We focused on miRNAs related to seed storability germination and identified zma-miR169o-5p, miR444a, zma-miR396c_L-1, and other differentially expressed miRNAs related to seed germination.

We identified a total of 218 known miRNAs and 448 putative miRNAs, most of which are 20–24 nt long, the 20.89% of sequences of 21nt, and the 54.32% of sequences of 24 identified as miRNAs were found, in line with the typical characteristics of Dicer enzyme cleavage of siRNAs in blueberry [41] and litchi [42]. We also found 25 known and 11 newly predicted miRNAs to be differentially expressed in the two maize inbred lines, indicating that miRNAs differ in their levels/regulation between seeds with distinct storability. In general, the identification of miRNAs in seed embryos during germination enriched the miRNA database of maize seeds.

In plants, miRNAs are involved in the regulation of growth and development, the synthesis of secondary metabolites, and crosstalk among distinct signal transduction pathways [43,44]. The previous studies confirmed that miR156 plays a significant role in seed development, dormancy and germination, and targeting and regulating squamous promoter binding proteins 10 (*SPL10*) and *SPL11* to prevent premature gene expression during early embryogenesis [45,46]. Interestingly, we found that *GRMZM2G399072*, a target gene negatively regulated by miR156e-5p, is a member of *AP2-ERF* family in our study of degradome sequencing and was not a previously reported *SPL* target gene. We hypothesized that miR156 may be involved in the regulation of many target genes, including *AP2-ERF* and *SPL*.

The high throughput sequencing identified the presence of zma-miR169o-5p, zma-miR390a-5p, zma-miR396c, zma-miR444a, zma-miR397b-p5, and novel-miR4 may play a regulatory role in seed storability. The qRT-PCR further confirmed opposite expression trends between these miRNAs and their target genes. We demonstrated the role of focused miRNAs, specifically miR444a and miR169o-5p, in the regulation of seed storability through 5 ‘RLM-RACE. According to previous studies, miR444 helps to regulate stress tolerance, plant growth, and development. Furthermore, MiR444a regulates the expression of *OsMADS27* and participates in the response of roots to nitrogen and phosphorus nutrients [47]; while tae-miR444a, along with its target gene *TaMADS57*, improves cold resistance in wheat [48]. In rice, *OsMADS27* was negatively regulated by miR444b.2 and its downstream target gene *NCED1*. The loss of *OsMADS27* function resulted in increased ABA content in the seeds of the mutant and delayed germination [49]. In our study, we found that miR444a was, respectively, up-regulated and down-regulated in the artificial aging treatment groups Dong156 and Dong237, and that the negatively regulated target gene *GRMZM2G399072* (AP2-ERF BBM2 subtype transcription factor) was down-regulated. These results indicate that miR444a inhibits seed germination by targeting AP2-ERF. Interestingly, *GRMZM2G399072* is also negatively regulated by miR156e-3p, indicating that multiple miRNAs regulate this gene through different pathways. MiR169/NF-YA may play an important role in seed morphogenesis, seed germination, root development, lateral organ formation, floral organ formation, stomatal formation, and stress response in plants [50,51,52]. A total of 10 and 14 *NF-YA* genes were identified by target gene prediction in Arabidopsis and maize, respectively [53,54]. Most of the sixteen target genes identified in rape miR169 family belong to the NF-YA family [55]. The 3′UTR of *NF-YA* in Arabidopsis and tobacco has a target site for miR169 cleavage, and miR169m/n/o has a cleavage-like effect on NF-YA3/9/2/12 [56]. The miR169/*NF-YA* module can be activated by endogenous signals or external environmental factors, whereby plants show enhanced characteristics to adapt to these changes [57]. To date, there have been no reports on miR169 expression during maize seed storability. In our study, we found that the miR169o-5p targeted gene *GRMZM2G165488* (*NF-YA11*) was down-regulated and *NF-YA11* was up-regulated in AA156VSAA237 comparison groups. However, further work is needed to understand how changes in the expression levels of miR169o-5p affect seed storability during the seed aging process.

Previous studies have shown that miRNAs negatively regulate the expression of target genes [58]. The miR169 family is plant-specific, and the target genes mainly include genes encoding plant NF-YA family transcription factors. In the late growth stage of *Arabidopsis*, *NF-YA2*/*NF-YA10* and miR169 subtypes can regulate the growth and development of the taproot and lateral root [50]. In our study, we found that miR169o-5p-targeted *GRMZM2G165488* (*NF-YA11*) was involved in the regulation of seed storability, with the down-regulated expression of miR169o-5p and the up-regulated expression of *NF-YA11*. The GO enrichment analysis of *NF-YA11* showed GO:0009793 (embryo development ends at seed dormancy), GO:0016602 (CCAAT binding factor complex), and GO:0048316 (seed development) are enriched terms. The end of seed dormancy is associated with abscisic acid (ABA) signal transduction through the involvement of several proteins that also participate in seed development [59]. Previous studies demonstrated that the overexpression of *NF-YA* can reduce the sensitivity of seeds to the ABA signal [51,59,60]. Therefore, miR169o-5p may regulate ABA signal transduction through the target gene *NF-YA11,* and thus affect seed development and morphogenesis to regulate seed storability. ABA crosstalk has often been associated with various plant stress responses [61,62,63]. Coincidentally, other studies have shown that miR169 family members miRC10 and *NF-YA11* were involved in regulating low nitrogen adaptation in maize [54]. Accordingly, we hypothesized that the target gene *NF-YA11* plays important functions and is not only involved in the regulation of seed germination but also in other important growth and developmental processes. The *AP2/ERF* transcription factor, as the hub of upstream signal and downstream functional gene connection in the signal transmission network, is deeply involved in stress response in rice [64]. For example, the miR1320-*OsERF096* module regulates cold tolerance by inhibiting the jasmonic acid-mediated cold signaling pathway [65]. The overexpression of *ZmEREB20* in Arabidopsis increased ABA sensitivity and affected seed germination under salt stress [66]. In maize, *AP2/ERF* is a large transcription factor family, with 292 potential members, 153 of which belong to the ERF subfamily [67], and which are extensively involved in various biological processes, such as growth, development, and stress response. We found that the negative regulation of miR444a and its target gene *AP2-ERF* may be associated with maize seed storability through the ethylene signaling pathway, even though the specific regulatory mechanisms need further investigation.

Recent studies indicated that a large number of miRNAs participate in complex regulatory networks to control stress responses and affect various developmental processes [68]. After artificial aging, two inbred lines differentially expressing zma-miR169o-5p, zma-miR390a-5p, zma-miR396c, zma-miR397b-p5, and novel-miR4 were found to be negatively correlated with seed storability, while the expression of miR444a was found to be positively correlated with seed storability by high-throughput sequencing. However, it is unclear how this mechanism was triggered and regulated, as is the connection between these miRNAs, both of which require further exploration into specific regulatory paths and mechanisms (Figure 7). These remain important questions that deserve future consideration. For example, studies on rice and *Arabidopsis* suggest that the growth regulatory factor (*GRF*) is a conserved target gene for miR396 in dicotyledons and monocotyledons [69,70], and that this miRNA participates in plant leaf growth and development probably by inhibiting *GRF* expression. Our small RNA sequencing and degradome sequencing results demonstrated that miR396 was down-regulated in the artificial aging comparison group. Combined with transcriptome analysis and functional annotation, three target genes with significant negative regulatory differences were identified as transcription factors (*MADS-box27*, *GRF14*, and *GRF3*), which were consistent with previous studies. This indicates that miR396 can negatively regulate the target gene *GRF* to participate in the regulation of seed development. Since *GRF* may regulate cell division, this inhibition of miR396 (which leads to and induction of *GRF*) may enhance cell division in seeds. Furthermore, combining the functional annotations of target genes and GO enrichment analysis, we found that zma-miR390a-5p and its *GRMZM2G134396* (*AMP-deaminase*) are involved in regulating seed storability through abscisic acid signaling and purine metabolism and that zma-miR397b and its target gene *GRMZM2G033230* (*bZIP108*) regulate seed development through salt and osmotic stress response. Beyond the target genes described above, it is certainly possible that other miRNAs and genes regulate seed storability. The continuous improvement of bioinformatic and molecular biology techniques should allow for the unravelling of the specific regulation mechanisms of miRNAs in seeds.

## 4. Materials and Methods

### 4.1. Plant Materials

The two inbred lines, Dong156 and Dong237, used in this study differ in their seed storability capacity (good and poor, respectively). The germination rate of the storage tolerant maize inbred line Dong156 remained above 90% for 8 years under natural storage conditions. In contrast, Dong237’s germination rate decreased to about 80% after one year of storage [71]. The seeds used in this study were harvested in 2021 and dried in the sun (with a moisture content below 14%). The samples were then stored at −20 °C in a low temperature freezer (DW-25W518, Qingdao Haier Co., Ltd. China) before further testing and analysis. The two inbred lines (Dong156 and Dong237) were cultivated by the Maize Research Institute of the Northeast Agricultural University (NEAU) in Harbin, China.

### 4.2. Seed-Accelerated Aging and Changes of Water Content Test during Seed Germination

The stored seeds were artificially aged by applying high temperature and humidity conditions (45 °C and 95% relative humidity) with different aging treatment levels [72]. After artificial aging, the seeds were dried at room temperature for 1–2 days, so that the water content of the seeds was restored to the original state. Standard germination tests were subsequently conducted. The seed surface was disinfected, and the germination rate was tested on moisture-proof filter paper. Germination was induced in a total of 50 seeds by placing them on moistened filter paper at 25 °C in the dark. Germination was considered as complete if/when the coleoptile grew more than 2 mm [40,73]. Each treatment was repeated three times for replicability purposes.

Dong156 and Dong237 seeds that had been artificially aged for 4 days were disinfected in 1.0% sodium hypochlorite solution and washed three times with deionized water. We followed the standard method for determining the water content of seeds [74]. Specifically, the fresh weight of the seeds was measured at different time points after germination by soaking the seeds in paper bed. The dry weight of seeds was measured after drying. Seeds germinated for 24 h after artificial aging for 4 days were immediately excised and frozen in liquid nitrogen, after which they were stored at −80 °C until further use [75].

### 4.3. RNA Extraction, Library Construction, and Sequencing

We used total RNA for Solexa sequencing and constructed six libraries: three sets from Dong156 after 24 h germination and 4 days of artificial seed aging; and the other three sets from Dong237 after 24 h after germination and 4 days of artificial seed aging. For every library, we combined 30 seeds per sample. Total RNAs was obtained for each sample using Trizol reagent (Tiangen, Beijing, China). Small RNA libraries were prepared with approximately 1 μg total RNA each sample, and we identified 18–30 nt RNA molecules using polyacrylamide gel electrophoresis (PAGE). The recovered small RNAs were then connected to the 3′ and 5′ end adapters, respectively, reverse transcribed and PCR amplified. After this, we used an Illumina hiseq2500 and single end reads (36 bp) for sequencing at the factory of lc-bio following manufacturer’s specifications (Hangzhou, China).

For degradome library construction, we used poly (A) RNA that was purified from maize embryo total RNA (20 µg) using poly-Toligo-attached magnetic beads. The potential slice targets of known and putative miRNAs were identified by high-throughput sequencing and the degradation products analyzed according to Cleveland 3.0 software package and ACGT101-miR (LC Sciences, Houston, Texas, USA).

Approximately 15μg of total RNA were used to prepare transcriptome libraries. Magnetic beads linked with Oligo (dT) were used to enrich eukaryotic mRNA after the total RNA was quantified. The extracted mRNA was fragmented at random using a fragmentation buffer lysate, and double-stranded cDNA was synthesized using random hexamer primers. T4 DNA polymerase and Klenow DNA polymerase were employed for terminal repair and 5′ olyadenylation, and PCR amplification was performed to obtain the final sequencing libraries. An Illumina Novaseq™ 6000 was used to sequence the original data after the library was quantified.

### 4.4. Sequence Data Analysis and Verification

We processed the raw reads using the following steps: deleted low quality labels, removed the labels with 5 primer contaminants, removed the label without 3 primers, removed the label without inserting it, removed the label with poly-A, and deleted labels less than 18 nt. After screening, we drew a clean label; compared the obtained sequence (Valid Reads) with the rRNAs, snoRNAs, tRNAs, and other sequences downloaded from the RFam database (http://rfam.janelia.org/ (accessed on 15 May 2022)); and found the small RNA without annotation information and compared it with the precursor and mature sequences selected in the miRBase (22.0) database. Finally, the sequenced sequence and precursor sequence were compared with the maize genome (ftp://ftp.ensemblgenomes.org/pub/release-43/plants/fasta/zea_mays/dn (accessed on 15 May 2022)). Compared with the reference genome, the sequenced sequence was divided into groups 1–4 according to the alignment results. Among them, only group 4 was not associated with the miRNA precursor sequences reported in miRBase. This part of the sequence was used to further search other miRNA and mRNA databases to predict whether it had a hairpin structure. In this case, the sequence type was putative miRNAs (Appendix A). We considered miRNAs and genes with |log2FC| (fold-change) higher than 0.58 and *p*-value < 0.05 as having altered expression.

We verified the validity of miRNA sequences and RNA-Seq by quantitative real-time PCR (qRT-PCR). We selected a total of 6 miRNAs of interest along with their target genes: miR169o-5p (*GRMZM2G165488*), miR444a (*GRMZM2G399072*), zma-miR390a-5p (*GRMZM2G134396*), zma-miR396c_L-1 (*GRMZM2G133568*/*GRMZM2G098594*/*GRMZM2G105335*), zma-miR397b-p5 (*GRMZM2G033230*), and novel-miR4 (*GRMZM2G009871*). Trizol kit (Quanshijin, Beijing, China) was used to separate total RNA (about 2 μg) from embryos germinated 24 h after artificial aging. We used 500 ng of RNA and the Transcript^®^ One Step gDNA Removal Kit MIX (Transgen Biotech, Beijing, China) to synthesize cDNA. The potential target genes for conserved miRNA and predicted target genes were identified from a maize genetic database. The specific primers for quantitative RT-PCR were designed with primer 5.0. We performed qRT-PCR using the TransStart Tip Green qPCR SuperMix fluorescent quantitative Kit (Genetically Modified Biotechnology Company, Beijing, China). The qRT-PCR with SYBR Green was performed on qTOWER3 G Touch (German Analytik Jena) real-time fluorescent quantitative gene amplifiers. In short, we cultured at 95 °C for 1 min, followed by 40 cycles of 94 °C for 10 s, 60 °C for 30 s, and 72 °C for 30 s. Finally, we performed a stage of 65 °C to 95 °C to confirm that primer dimers and no multiple products were present. The U6 was used as an endogenous control. All samples were repeated three times for replicability purposes. The obtained qRT-PCR data were used for analysis. All primers are listed in Appendix A.

Total RNA was extracted from seed embryos 24 h after artificial aging germination. For cDNA production, the starting material (3 to 5 µg of columnar clean RNA) was processed using the FirstChoice™ RLM-RACE Kit (Abion, Austin, TX, USA), with alkaline phosphatase and acid pyrophosphatase steps excluded according to the manufacturer’s instructions. After the adapter was connected, gene specific primers designed for each target were used according to the coding sequence reported in MaizeGDB. Two gene specific primers (GSP and NGSP) were designed by primer premier 5.0 software (Appendix A). We performed RACE according to the instructions of the GeneRacer^TM^ Kit: RLM-RACE (Invitrogen, Waltham, MA, USA). After the reaction, 1% agarose gel was used to detect the PCR product. The target band was recovered, connected with the T-vector, and transformed into *E. coli* for susceptibility, screening positive clones, and sequencing. The sequencing results were compared to the corresponding target gene sequences to determine the shear sites of each cloned sequence.

### 4.5. Transcriptome—Small RNA—The Degradome Combined Analysis

The relationship between miRNA and target gene was obtained by degradome sequencing, and the expression profile of miRNA and target gene in the difference comparison group was integrated to obtain the negative regulatory relationship between miRNA and target genes. The network regulation analysis of the focused miRNA and target genes was performed with Cytoscape 3.2.0. *p*-value ≤ 0.05 was used as the threshold, and the main biological functions of miRNA-target gene pairs were determined by functional significance enrichment analysis.

## 5. Conclusions

We defined the key sampling time by measuring the germination percentage of two inbred lines at different aging times, as well as the water content of seeds during germination. Through miRNA sequencing and degradation group sequencing, we identified 218 known miRNAs and 448 new predicted miRNAs in two maize inbred lines (Dong156 and Dong237). Among them, twenty-five known and eleven newly predicted miRNAs were differentially expressed in two maize inbred lines, and six pairs of differentially expressed miRNAs and their negative regulatory target genes related to seed storability were screened and confirmed by transcriptome sequencing and qRT-PCR. GO enrichment analysis showed that the miRNA and target genes regulated seed storability through the ethylene activation signal pathway, hormone synthesis and signal transduction, and plant organ morphogenesis. The cleavage sites of miRNA and target genes were verified by 5′RLM-RACE and qRT-PCR methods. These findings provide valuable information to better understand how miRNAs contribute to maize seed storability.

## Figures and Tables

**Figure 1 ijms-23-12339-f001:**
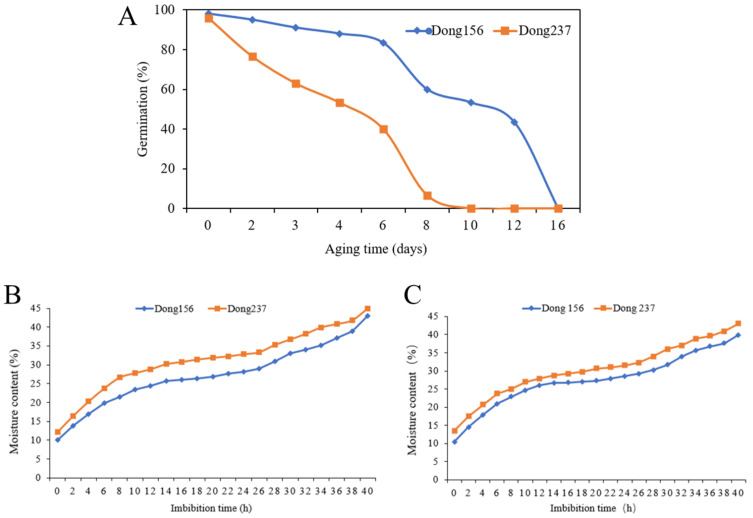
Changes of germination percentage and water content of two inbred lines Dong 156 and Dong 237. (**A**) The change in Dong156 and Dong237 seed germination during accelerated aging. The seeds were aged under 45 °C and 95% RH. The error bars indicate ± SE (*n* = 3), (**B**) the moisture content change curves of Dong156 and Dong237 of the control group, (**C**) the moisture content change curves of in the fourth day artificial aging treatment group Dong156 and Dong237.

**Figure 2 ijms-23-12339-f002:**
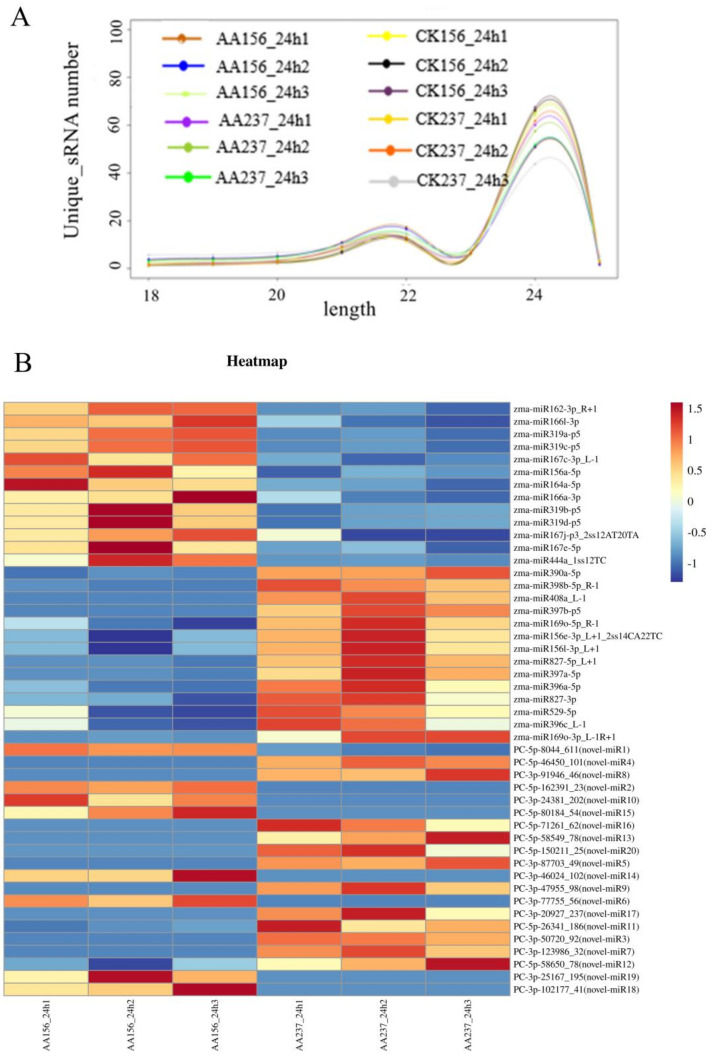
Summary of miRNA sizes and differentially expressed miRNAs and the number of identified known miRNAs. (**A**) The size distribution of unique sRNAs. (**B**) Differential expression heat maps of known miRNAs and putative miRNA s in maize. R+n means that there are n more bases on the right end of the miRNA included in the miRBase, R−n means that there are n fewer bases on the right end of the miRNA included in the miRBase, L+n means that there are n more bases on the left end of the miRNA included in the miRBase, L+n means that there are n more bases on the left end of the miRNA included in the miRBase, 2ss12AT20TA means that the 12th base T is replaced by A (ss, substitution), and the 20th base T is replaced by A, a total of 2. New miRNAs are labeled with PC (Predicted Candidate).

**Figure 3 ijms-23-12339-f003:**
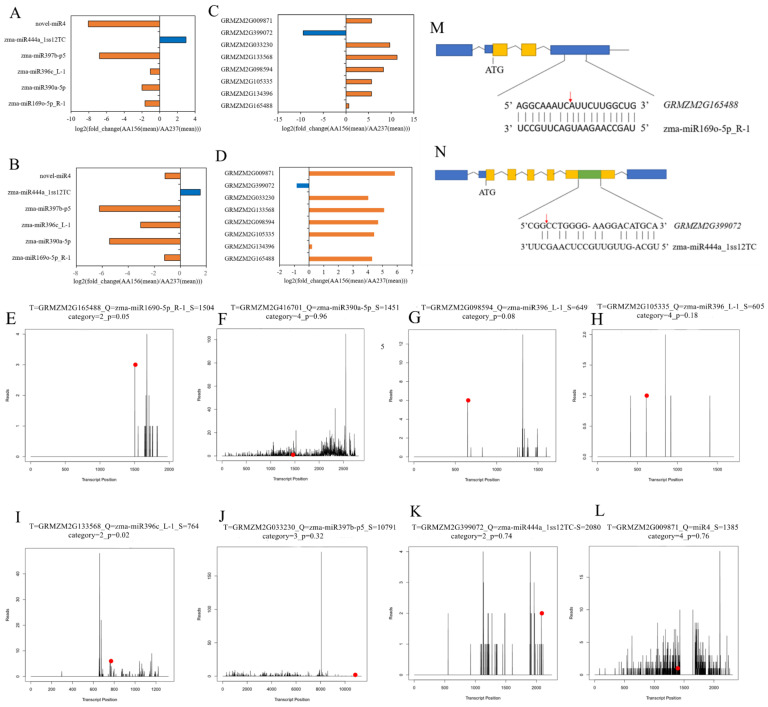
The miRNA, target, and degradation product abundances by RNA seq and validation by alternative methods. (**A**) high throughput sequencing to verify miRNA, the orange bars indicate the up-regulated miRNA and the blue bars indicate the down-regulated miRNA. (**B**) qRT-PCR to verify miRNA, the orange bars indicate the up-regulated miRNA and the blue bars indicate the down-regulated miRNA. (**C**) high throughput sequencing to verify target genes, the orange bars indicate the up-regulated target genes and the blue bars indicate the down-regulated target genes. (**D**) qRT-PCR to verify target genes, the orange bars indicate the up-regulated target genes and the blue bars indicate the down-regulated target genes. (**E**–**L**) the relative abundance of zma-miR169o-5p, zma-miR390a-5p, zma-miR396c_L-1, zma-miR397b-p5, zma-miR444, and novel-miR4 and their target genes. The red dot is the same position of the predicted target gene and the degradation site obtained from the degradome sequencing, and the black line is the actual degradation site measured in the degradome sequencing. (**M**) zma-miR169o-5p_R-1 acts on the cleavage site of the target gene *GRMZM2G165488* were confirmed by 5′RLM-RACE. (**N**) zma-miR444a_1ss12TC acts on the cleavage site of the target gene *GRMZM2G399072* were confirmed by 5′RLM-RACE. The yellow bars represent the exon part of the gene, the green bars represent the cutting site part of the target gene, the black broken line represents the intron part of the gene, and the blue bars represent other structures of the gene.

**Figure 4 ijms-23-12339-f004:**
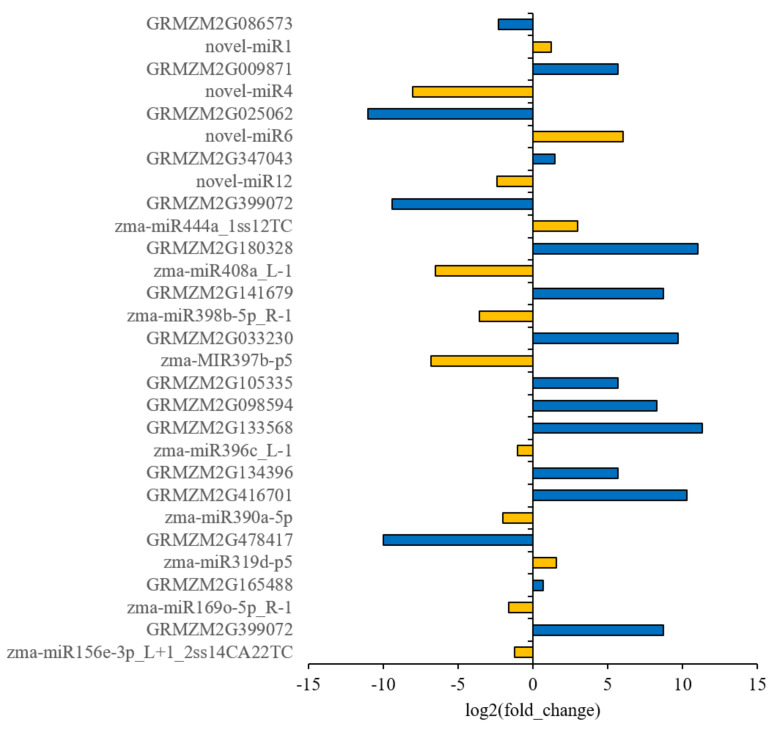
Differential expression of the known and putative miRNAs and target genes in maize. FC, fold change. Orange bars represent expression of miRNAs; blue bars represent expression of target genes.

**Figure 5 ijms-23-12339-f005:**
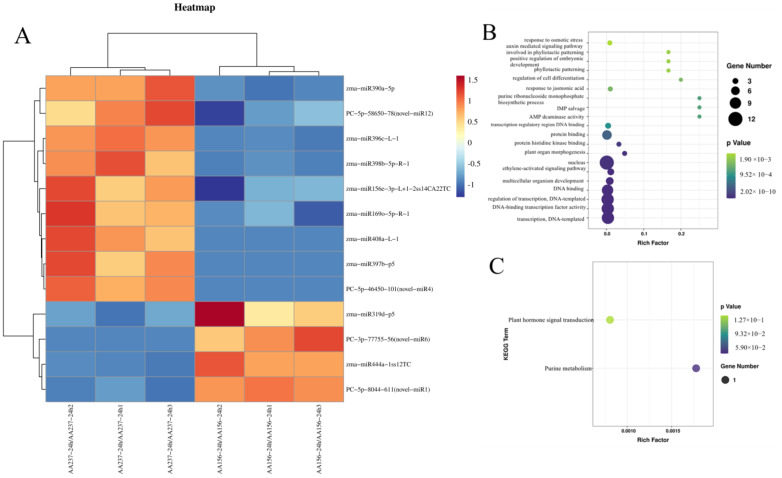
Analysis of the targets of identified miRNAs in maize seed storability. (**A**) Hierarchical clustering of DEGs expression. (**B**) Profile of GO analysis of the targets of identified responsive miRNAs in maize seed storability. (**C**) Profile of KEGG analysis of the targets of identified responsive miRNAs in maize seed storability.

**Figure 6 ijms-23-12339-f006:**
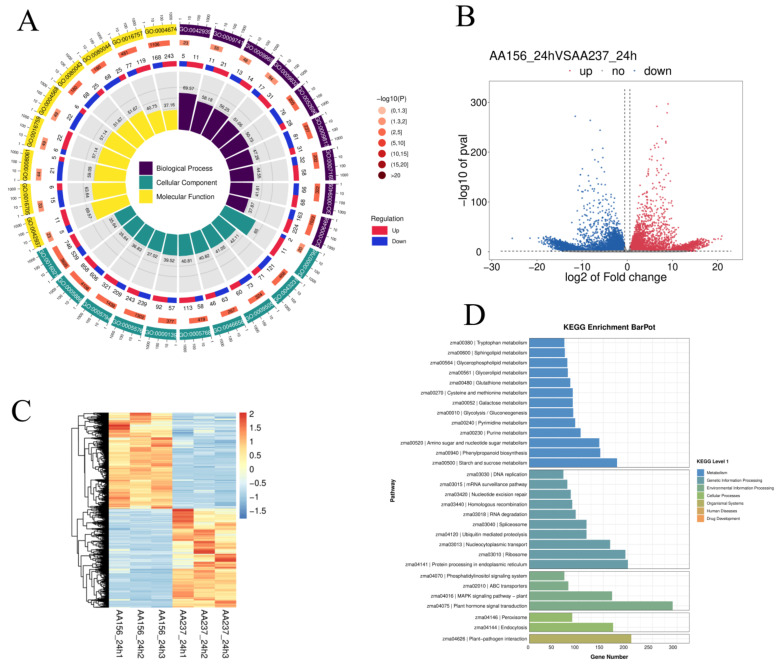
Analysis of differentially expressed genes in transcriptome. (**A**) Gene ontology LoopCircos of DEGs. (**B**) The volcano map of differentially expressed genes. (**C**) Hierarchical clustering of DEGs expression. (**D**) KEGG enrichment analysis of DEGs.

**Figure 7 ijms-23-12339-f007:**
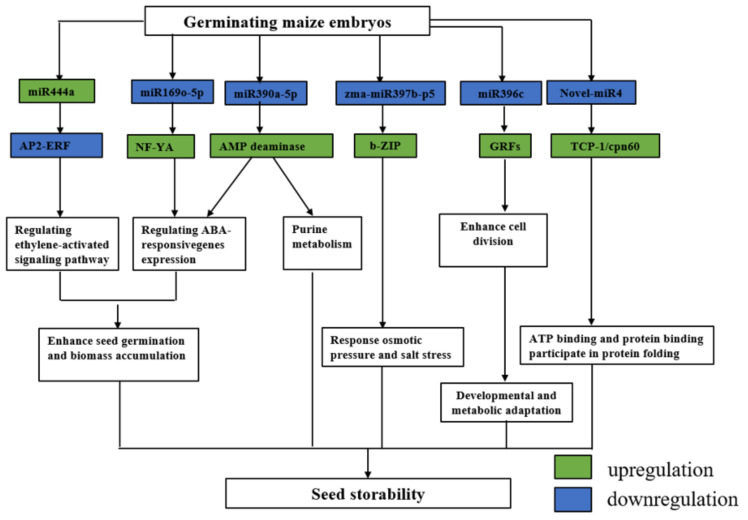
Possible microRNAs-dependent regulatory pathways that participate in seed storability during maize germination.

**Table 1 ijms-23-12339-t001:** A total of 27 known miRNAs were differentially expressed.

miR_Name	miR_Seq	Up/Down	Fold_Change (AA156_24h (Mean)/AA237_24h (Mean))	log2 (Fold_Change)	*p* Value (*t*_Test)	AA237_24h (Mean)	AA156_24h (Mean)
zma-miR156a-5p	TGACAGAAGAGAGTGAGCAC	up	4.13	2.04	0.0203	366	1510
zma-miR156e-3p_L+1_2ss14CA22TC	TGCTCACTGCTCTATCTGTCACC	down	0.42	−1.24	0.0132	220	93
zma-miR156l-3p_L+1	TGCTCACTGCTCTATCTGTCACC	down	0.42	−1.24	0.0132	220	93
zma-miR162-3p_R+1	TCGATAAACCTCTGCATCCAG	up	12.92	3.69	0.0039	3	44
zma-miR164a-5p	TGGAGAAGCAGGGCACGTGCA	up	4.96	2.31	0.0274	23	115
zma-miR166a-3p	TCGGACCAGGCTTCATTCCCC	up	3.83	1.94	0.0442	24,545	94,027
zma-miR166l-3p	TCGGACCAGGCTTCATTCCTC	up	3.95	1.98	0.0046	412	1625
zma-miR167c-3p_L-1	ATCATGCTGTGGCAGCCTCACT	up	5.18	2.37	0.0095	28	146
zma-miR167e-5p	TGAAGCTGCCAGCATGATCTG	up	2.70	1.43	0.0459	54	147
zma-miR167j-p3_2ss12AT20TA	TCTGAAGAAAGTGTTGGCTATC	up	4.71	2.23	0.0451	2	7
zma-miR169o-3p_L-1R+1	GCAGGTCTTCTTGGCTAGCC	down	0.02	−5.96	0.0459	153	2
zma-miR169o-5p_R-1	TAGCCAAGAATGACTTGCCT	down	0.32	−1.64	0.0111	1327	427
zma-miR319a-p5	TAGCTGCCGACTCATCCATTCA	up	6.62	2.73	0.0068	19	128
zma-miR319b-p5	AGCTGCCGACTCATTCACCCA	up	2.94	1.56	0.0450	241	708
zma-miR319c-p5	TAGCTGCCGACTCATCCATTCA	up	6.62	2.73	0.0068	19	128
zma-miR319d-p5	AGCTGCCGACTCATTCACCCA	up	2.94	1.56	0.0450	241	708
zma-miR390a-5p	AAGCTCAGGAGGGATAGCGCC	down	0.25	−1.99	0.0016	237	60
zma-miR396a-5p	TTCCACAGCTTTCTTGAACTG	down	0.21	−2.26	0.0248	1603	334
zma-miR396c_L-1	TCCACAGGCTTTCTTGAACTG	down	0.48	−1.05	0.0455	519	251
zma-miR397a-5p	TCATTGAGCGCAGCGTTGATG	down	0.03	−4.87	0.0233	42	1
zma-miR397b-p5	TTGAGCGCAGCGTTGATGAGC	down	0.01	−6.79	0.0103	40,574	366
zma-miR398b-5p_R-1	GGGGCGGACTGGGAACACAT	down	0.08	−3.56	0.0063	340	29
zma-miR408a_L-1	TGCACTGCCTCTTCCCTGGC	down	0.01	−6.51	0.0083	9270	102
zma-miR444a_1ss12TC	TGCAGTTGTTGCCTCAAGCTT	up	7.87	2.98	0.0484	820	6457
zma-miR529-5p	AGAAGAGAGAGAGTACAGCCT	down	0.60	−0.73	0.0453	628	379
zma-miR827-3p	TTAGATGACCATCAGCAAACA	down	0.25	−2.00	0.0306	10,114	2520
zma-miR827-5p_L+1	TTTTGTTGGTGGTCATTTAACC	down	0.10	−3.36	0.0154	2511	244

**Table 2 ijms-23-12339-t002:** Transcriptome–small RNA–degradation group combined analysis.

miRNA	log2 (Fold_Change)	Up/Down	Transcript	GnenID	log2 (Fold_Change)	Description	Up/Down	Focus on GO	KEGG
zma-miR169o-5p_R-1	−1.63707	down	XM_008682881.3	GRMZM2G165488	0.68	Nuclear transcription factor Y subunit A-10	up	GO:0009793 (embryo development ending in seed dormancy); GO:0048316 (seed development)	NA
zma-miR390a-5p	−1.98813	down	NM_001320234.1	GRMZM2G134396	5.682455	uncharacterized protein LOC100275036	up	GO:0003876 (AMP deaminase activity) GO:0009737 (response to abscisic acid);GO:0009793 (embryo development ending in seed dormancy)	NA
zma-miR396c_L-1	−1.04713	down	XM_008649043.2	GRMZM2G098594	8.267418	putative growth-regulating factor 14	up	GO:0008757 (S-adenosylmethionine-dependent methyltransferase activity)	NA
			XM_008678196.2	GRMZM2G105335	5.682455	putative growth-regulating factor 3	up	GO:0009739 (response to gibberellin)	NA
			XM_020552600.2	GRMZM2G133568	11.31	MADS-box transcription factor 27 isoform X1	up	GO:0008134 (transcription factor binding)	NA
zma-MIR397b-p5	-6.79	down	XM_020541848.1	GRMZM2G033230	9.73	putative bZIP transcription factor superfamily protein	up	GO:0006970 (response to osmotic stress);	NA
zma-miR444a_1ss12TC	2.976548	up	XM_008653282.2	GRMZM2G399072	−9.41933	AP2-like ethylene-responsive transcription factor BBM2 isoform X2	down	GO:0009873 (ethylene-activated signaling pathway)	NA
novel-miR4	−8.05	down	XR_002268591.2	GRMZM2G009871	5.6825732	putative TCP-1/cpn60 chaperonin family protein	up	GO:0044183 (protein binding involved in protein folding)	NA

## Data Availability

The microRNA, degradome, and transcriptome sequencing data have been deposited in the Sequence Read Archive (SRA) at the National Center for Biotechnology Information (NCBI) under the accession number PRJNA877842, PRJNA878616, and PRJNA877700; their Reviewer links are https://dataview.ncbi.nlm.nih.gov/object/PRJNA877842, https://dataview.ncbi.nlm.nih.gov/object/PRJNA878616 and https://dataview.ncbi.nlm.nih.gov/object/PRJNA877700 (accessed on 7 September 2022.)

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
