# Peer review of "Identification of miRNAs Mediating Seed Storability of Maize during Germination Stage by High-Throughput Sequencing, Transcriptome and Degradome Sequencing"

_ijms, 2022, doi:10.3390/ijms232012339_

Round 1

Reviewer 1 Report

The authors present the analysis of differential miRNA accumulation (24 h upon imbibition) between two maize inbred lines with distinct germination ability after four days of artificial aging. They correlate such accumulation with degradome and transcriptome data to find out the regulation on specific targets for these miRNAs that could differ between lines and hence explain their differential storability response.

The manuscript is quite unclear in presenting the data and I could not appreciate a solid contribution of the findings. There is no demonstration of the involvement of miRNAs and targets in seed storability Maybe it is more appropriate to say: ... to identify the miRNAs and targets that differ in their levels/regulation between seeds with distinct storability. Importantly, there is no public repository of any raw data of sRNA sequencing, transcriptomes or degradomes presented in the paper, nor as supplementary information. 

There is a previous paper entitled “Identification of miRNAs and Their Target Genes Associated with Sweet Corn Seed Vigor by Combined Small RNA and Degradome Sequencing” by Gong et al., published in 2015, cited by the authors. The authors should clearly indicate, similarities or differences in their approach with respect to this previous report. What is the contribution of their findings. Where are more abundant the miRNAs and the targets? How was the comparison performed? How does the correlation with storability behaves? Worse or better according to which miRNA/target over- or down-accumulation and which inbred maize line?

Regarding known miRNA family members shown in Figure 4, if the authors are considering both, 3p and 5p versions for each miRNA, they should mention this. It would be better to include only the functional miRNAs in this Figure.

Figures 7, 8 and 9 have poor resolution, and one could not distinguish the letters for GO groups, KEGG classification, accessions, etc. One should go to the supplementary tables to see the identity of labels. Figure explanations in legends are practically absent. The GO term explanation is not found even in the supplementary table. The confirmation of miRNA targets is relevant for the paper, but the Figure contains no demonstration. When looking for this in Supplementary Figure S1, I could not figure out what the authors were showing. In addition, cleavage site is at one extreme of the miRNA:target pairing and complementarity is very poor. This is very odd; cleavage usually takes place at 10-11 nt and there should be better complementarity between miRNA and targets for cleavage to take place.

There are many conceptual errors throughout the manuscript creating lot of confusion. Just to show some examples:

Line 49: unclear phrase: “seed storage occurs during cell metabolism”. Is the seed storage accompanied by some detrimental cell metabolism? Or: Does the previous metabolic state during seed desiccation affect storability?

Lines 101-103: “…these negatively regulated miRNAs”: where and/or when are the miRNAs negatively regulated? Which miRNAs are negatively regulated and how do they impact on the ethylene signaling pathway?

Line 142 – 143. Authors state: “We found that most sequences in the six libraries were 24 nt long, which is the length of mature miRNAs in most plants (between 20-24nt; Figure 2A).” 1) The common length of miRNAs is 21 nt, but not 24 nt. In a library, there are many small RNAs that are not miRNAs, such as siRNAs which are 24 nt long. Just a few miRNAs in plants are 24 nt long. 2) The Figure authors refer to is not Figure 2, but Figure 3 in their Figures at the end of manuscript.

Line 144 – 147. Authors state: “A unique sequence with a length of 18-25 nucleotides 144 was mapped to the miRBase 20.0 (http://microrna.sanger.ac.uk/), and both the known and 145 novel miRNA sequences from the 3' (miRNA name 3P) and 5' (miRNA name 5p) ends were identified by explosion (Figure 2B).” The whole sentence is rare. Current release of miRBase is 22.1; even in 2019, there was already miRbase 21.0. The authors should map their sequences to a more recent database, same as for the maize genome database. What is a unique sequence? How do they map “novel” miRNAs to miRBase if they are novel? I could not appreciate the description of any solid criteria to present these novel miRNAs. The only criteria the authors mention in Methods is the formation of stable hairpin in the mapping region of the genome, but no evidence is shown in Figures or Supplementary. Even if hairpin was found, the authors should show proportion of reads mapping to this structure to evidence specific production of a miRNA.

Author Response

“请参阅附件。

Reviewer 2 Report

Comments to manuscript PlaBio-2022-07-0273-RP “Insights into microRNA-mediated interaction and regulation of different metabolites in tomato" in attachment 

  •  

Reviewer 3 Report

The manuscript "ijms-1929358" is exploring an important area in seed storage. The experiment is well planned and executed. The manuscript is also well written except minor corrections.

L49: Needs rewording

L112: ...at high temperature and humidity (45oC, 95% RH)...

L447: snrnas or snRNA's?

L467 and L482: why only 24h artificial aging?

L484: rlrace or RLM RACE?

L489: ...SMARTtm RACE?...

L501: ... seed storability during germination...? Don't understand the phrase.

L503: ...seed storage tolerance...?  Don't understand the phrase

L504-L506: Needs rewording

Fig 6,7, 8, and 9: font size not visible for readers. Fig. 8 and 9 graph is not visible.

Round 2

Reviewer 1 Report

The second version of the manuscript has been improved in clarity and description of details that allow to appreciate the contribution of the work. However, I still consider there are relevant changes to make in some Figures and result descriptions, before accepting it for publication.  

V2.1 Lines 102-105: Regarding Response 2, I recommend the authors shortly state novel contribution of their work after citing the preceding Gong et al., paper. For example, “… maize seed storability (40). Here we adopted the method of high temperature and humidity artificial aging in two maize inbred lines with significant differences in seed storability and performed global RNA analysis at 24 h of seed imbibition after 4 days of aging. In the analysis we included transcriptome, in addition to miRNA and degradome sequencing, to identify differentially expressed miRNAs and their target genes more accurately.”

V2.2 Line 150: “Unique sequences …. were …” instead of “A unique sequence … was …” Regarding Response 3, many of the sequenced miRNAs could match to more than one precursor (at least for known miRNAs) and you could not assign a particular family member (a, b, c, d…) if they share the same mature sequence. You could look for pre-miRNA expression at the transcriptome data, but these would not tell the origin of mature miRNA sequenced in the small RNA library. For example, since mature miR390a-5p and miR390b-5p share the same mature sequence, I could not understand the basis of indicating miR390a-5p on Figure 3 A and B.

V2.3 Comments on Figures/Tables

Legends are still lacking explanations of Figures

Figure 2A. I shall insist that these data are not representing miRNAs, but small RNAs. According to miRbase or miREN, almost none of the reported (known) miRNAs in maize is 24 nt long. Therefore, the y axis should refer unique sRNA number, instead of unique miRNA nums. Just considering the 24 nt bar, would you be reporting near 1000 unique miRNA sequences??? Please also correct within the text (Lines 150-154).

 Figure 2B: Please indicate the meaning of R+/-1, L+/-1, 2ss12AT20TA and other letter/number additions to miRNA names. Also indicate what the right scale refers to. Are you comparing AA156_24h1 to AA237_24h1 or how the comparison was made? Are you indicating log2FC?

 Figure 2C: This panel could be omitted since it is still confusing. Are these results from the analysis? What I see is that there are a number of miRNAs belonging to the same family as reported in miRbase. From small RNA sequencing it is not possible to distinguish between members of the same family if the mature miRNA shares the same sequence with other members. In addition, the graph is not representing data from the sequencing, just the distribution of members within each miRNA family.

 Figure 3. I suggest rephrasing the Figure title to include all information shown in the Figure. For example: “miRNA, target, and degradation product abundances by RNA seq and validation by alternative methods.” Regarding panels E-L, I assume they represent degradome data for each miRNA target indicated on top. However, there is no explanation about the red dot meaning. If it points towards the miRNA targeted position, or something else please clarify in the legend. The miRNA targeted position should be indicated in these panels. It should be also clearly presented in lines 224-226.

Table S2: the p-value column only displays #####

Table S7: Brief explanation is required to understand each group of miRNA-mRNA-GO, there are many genes indicated for each relation, what does it mean? Are they also targets of the same miRNA? According to what criterion? Degradome maybe?

Table S8: I understand that comparisons have been made to miRNA precursors in miRbase in the procedure. However, when grouping unique sequences in the small RNA libraries they could belong to more than one precursor. Therefore, designation to a particular precursor is not possible for many miRNAs (i.e. miR156, miR390…). Please, include how the designation to a particular precursor of known mature miRNA has been performed.  

Reviewer 2 Report

The authors have attended all observations, regarding the methodology they have no way to change the analysis using shortstack. The introduction, discussion and conclusions have been improved. They have also significantly improved the structure and order of the figures.

Line 112. genesis ,there , delete space before ,

Line 120. (45 ° C , 95% RH), delete space before ,

Line 304. [40], Additionally…change , by .

Line 305 MYB33  MYB101 in cursive

Round 3

Reviewer 1 Report

I agree with the current version of the manuscript.